# Use of multi-color flow cytometry for canine immune cell characterization in cancer

**Maciej Parys**[1]*, **Spela Bavcar**[1¤], **Richard J. Mellanby**[1], **David Argyle**[1], **Takanori Kitamura**[1,2]*

**1** Division of Infection and Immunity, Veterinary Clinical Sciences, Royal (Dick) School of Veterinary Studies and The Roslin Institute, University of Edinburgh, Midlothian, United Kingdom, **2** MRC Centre for Reproductive Health, Queen's Medical Research Institute, University of Edinburgh, Edinburgh, United Kingdom

¤ Current address: Small Animal Specialist Hospital, North Ryde, New South Wales, Australia
* Maciej.Parys@ed.ac.uk (MP); tkitamur@exseed.ed.ac.uk (TK)

**Data Availability Statement:** All relevant data are within the manuscript and its Supporting Information files. The raw data of flow cytometry

## Abstract

Although immunotherapy is becoming a standard approach of human cancer treatment, only a small but critical fraction of patients responds to the therapy. It is therefore required to determine the sub-populations of patients who will respond to immunotherapies along with developing novel strategies to improve efficacy of anti-tumor immune reactions. Current development of novel immunotherapies relies heavily on mouse models of cancer. These models are important for better understanding of mechanisms behind tumor immune escape and investigation of novel strategies to overcome it. Nevertheless, the murine models do not necessarily represent the complexity of spontaneously occurring cancers in humans. Dogs spontaneously develop a wide range of cancer types with an intact immune system under similar environment and exposure to humans, which can serve as translational models in cancer immunotherapy research. To date though, there is still a relatively limited amount of information regarding immune cell profiles in canine cancers. One possible reason could be that there are hardly any established methods to isolate and simultaneously detect a range of immune cell types in neoplastic tissues. To date only a single manuscript describes characterization of immune cells in canine tumour tissues, concentrating solely on T-cells. Here we describe a protocol for multi-color flow cytometry to distinguish immune cell types in blood, lymph nodes, and neoplastic tissues from dogs with cancer. Our results demonstrate that a 9-color flow cytometry panel enables characterization of different cell subpopulations including myeloid cells. We also show that the panel allows detection of minor/aberrant subsets within a mixed population of cells in various neoplastic samples including blood, lymph node and solid tumors. To our knowledge, this is the first simultaneous immune cell detection panel applicable for solid tumors in dogs. This multi-color flow cytometry panel has the potential to inform future basic research focusing on immune cell functions in translational canine cancer models.

are accessible via http://flowrepository.org/id/FR-FCM-Z5Q7.

**Funding:** This work was supported by Canine Welfare Grant from Dogs Trust UK (Project ID 8841383) - awarded to MP and TK. The funders had no role in study design, data collection and analysis, decision to publish, or preparation of manuscript.

**Competing interests:** The authors have declared that no competing interests exist.

## Introduction

Immunotherapies including the use of checkpoint inhibitors and transfer of engineered T cells are emerging therapeutic approaches that have shown great promise in certain types of human cancer, in particular melanoma and lymphoma [1]. However, their efficacies are limited to a fraction of patients likely due to suppressive effects of tumor microenvironment on immune system [2, 3]. Therefore, a deeper understanding of immune cell phenotype and functionality within tumors before and during immunotherapy is required. To this end, mouse models of cancer are the strongest tools besides human clinical samples, which have been contributing to understand immune reaction in tumours and to reveal efficacy and mechanism of action of novel immunotherapies. However, mouse models have some disadvantages when utilized for translational research, e.g., mice rarely develop spontaneous cancer, laboratory mice are genetically homogeneous and housed in carefully controlled environment, severe immune deficiency is required to transplant human cancer materials, and biodistribution of compounds is different from humans due to their small size. In contrast to mice, dogs develop cancers spontaneously with fully functional immune system, have diverse genetic backgrounds and microbiomes, and are exposed to environments similar to human. Moreover, cancer is one of the most common causes of death in dogs, similar to humans. For example, within certain canine breeds such as Rottweilers or Bernese Mountain Dogs, close to 50% of animals die due to neoplasia [4]. Thus, dogs are a unique model to study tumor microenvironment and next generation immunotherapies that link data from mouse models and human clinical samples. Nevertheless, to date the studies characterizing tumor infiltrating immune cells are scarce since most studies have concentrated on immunohistochemical investigations of a single type of cells. Although flow cytometry is now commonly used for diagnosis of lymphoma/leukemia in dogs, the majority of currently published protocols utilize single- to four-color flow cytometry panel for phenotyping of immune cells [5–8]. This approach allows to identify the dominant cell population in cancer samples, which is necessary for diagnosis. However, it does not allow to distinguish different immune cell types (e.g., monocytes, neutrophils, CD4$^+$ T, CD8$^+$ T, and B cells) or detect minor subpopulations (e.g., monocyte subsets) in cancer that can play an important role in tumor progression and prognosis. Additionally, in some cases of lymphoma, neoplastic immune cells are reported to aberrantly express immune cell markers. For example, cells in the lymph node of a subset of canine patients co-express B-cell, T-cell and/or myeloid markers [5]. This makes it more difficult to distinguish immune cell types in cancer-bearing animals properly. A multi-color flow cytometry is a powerful approach to identify the aberrant expression of immune cell markers in neoplastic cells and to distinguish different types of immune cells infiltrating into the neoplastic tissues. Moreover, thorough characterization of immune cell types using a multi-color panel rather than a single to triple stain allows to determine precise CD4$^+$ T /CD8$^+$ T ratio or neutrophil/T cell ratio, which have been described to be important prognostic marker in human and canine cancers [9–11]. However, limited option of fluorochrome conjugation for commercially available anti-canine antibodies is so far a bottle neck to use extended marker panels. Moreover, methods for simultaneous detection of lymphoid and myeloid cells within neoplastic tissues has not been reported. A recent study described a 5-color flow cytometry panel that enables to reveal differences between T cell subsets in the blood and those in melanoma tissues [12]. However, this panel is designed to characterize specific T cell subsets, rather than broad categories of immune cell subsets. Herein we describe a method for multi-color flow cytometry using a minimum of nine antibodies to detect canine immune cells and validate applicability of this panel in various canine neoplastic samples including blood, lymph node, and solid tumor. To date a majority of studies base their insights into tumor microenvironment on immuno-histochemical studies, or small, limited

panels targeting specific types of cells [12–15]. The approach described in this study enables the characterization of multiple immune cell subsets rather T cells only in one tube, which will help to build new insights into tumor microenvironment in translational cross-species studies.

## Materials and methods

The protocol described in this peer-reviewed article is published on protocol https://dx.doi.org/10.17504/protocols.io.kqdg3p45el25/v1 and is included for printing as supporting information file 1 with this article (**S1 File**).

### Samples

The residual blood and tissue samples were collected at the Royal (Dick) School of Veterinary Studies, Hospital for Small Animals with a written consent from owners of dogs suffering from cancer or inflammatory disease. Each animal had appropriate analgesia provided, which was selected by clinician in charge and depended on the type of procedure/condition.

### Ethics declarations

This study has been reviewed and approved by the R(D)SVS Veterinary Ethical Review Committee (VERC reference number: 99.19).

## Expected results

### Development of 9-color flow cytometry panel

To distinguish T cells, B cells, neutrophils, and monocytes by flow cytometry, we first listed antibodies that are reported to detect marker proteins for these cell types in a dog. Since variation of fluorochrome conjugated to canine specific antibodies is still limited, we selected a minimum of nine antibodies that are required to detect the major immune cell types and developed a panel without overlapping the color of conjugates (**Table 1**).

To validate this panel, we analyzed peripheral blood samples collected from dogs presenting with clinical suspicion of lymphoma, based on peripheral lymphadenopathy (n = 12), concurrently to the fine-needle aspirate of the lymph node sample.

After gating a live single cell population, CD45 positive leukocytes were separated into T cells (CD5$^+$CD21$^-$), B cells (CD5$^-$CD21$^+$), and other leukocytes (CD5$^-$CD21$^-$) (**Fig 1A**). The

**Table 1. List of antibodies used in this study.**

| Antigen | Clone | Fluorochrome | Host* | Isotype | Target* | Source | Volume per 2.5x10$^5$ cells |
|---|---|---|---|---|---|---|---|
| CD45 | YKIX716.13 | PE | R | IgG2b | D | Bio-Rad | 1.25 μL (1.25 μg) |
| CD5 | YKIX322.3 | FITC | R | IgG2a | D | Bio-Rad | 1.25 μL (1.25 μg) |
| CD21 | CA2.1D6 | APC | M | IgG1 | D | Bio-Rad | 1.25 μL (1.25 μg) |
| CD4 | YKIX302.9 | PE/Cy7 | R | IgG2a | D | Bio-Rad | 1.25 μL (1.25 μg) |
| CD8 | YCATE55.9 | P.Blu | R | IgG1 | D | Bio-Rad | 1.25 μL (1.25 μg) |
| CD11b | M1/70 | PerCP/Cy5.5 | R | IgG2b | M/H$^#$ | Biolegend | 0.50 μL (0.1μg) |
| Neutrophil | CADO48A | Purified$^†$ | M | IgG1 | D | Kingfisher Biotech | 0.25 μL (0.25μg) |
| CD14 | TÜK4 | AF700 | M | IgG2a | H$^#$ | eBioscience | 1.00 μL(0.05μg) |
| MHCII | L243 | BV650 | M | IgG2a | H/D | Biolegend | 2.50 μL (0.25 ug) |

*R, rat; M, mouse; D, dog; H, human.

$^#$Cross reactivity in dog is reported (Reference # [16]).

$^†$Conjugated with APC/Cy7 using LYNX Rapid APC-Cy7 Antibody Conjugation Kit (Bio-Rad).

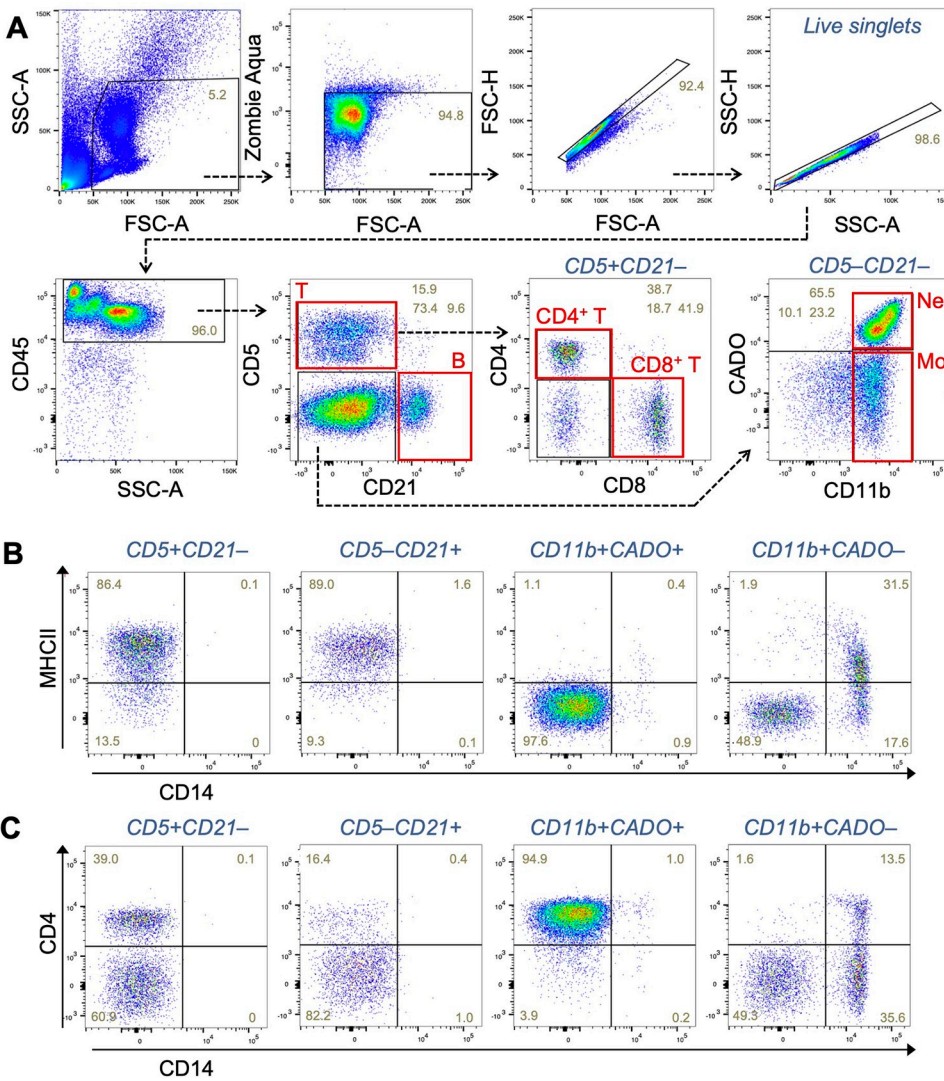

**Fig 1. Development of a flow cytometry panel using 9 different immune cell markers. (A)** Representative dot plots showing gating strategy to detect major immune cell types in peripheral blood of dogs with enlarged lymph node and diagnosed with B-cell lymphoma. T, T cells (CD5$^+$CD21$^-$); B, B cells (CD5$^-$CD21$^+$); Neu, neutrophils (CD5$^-$CD21$^-$CD11b$^+$CADO$^+$); Mo, monocytes (CD5$^-$CD21$^-$CD11b$^+$CADO$^-$). **(B, C)** Representative dot plots showing the expression of CD14 and MHC class II (B) or CD4 (C) in each immune cell type gated in A.

CD5$^+$CD21$^-$ cells were further gated to helper (CD4$^+$CD8$^-$) and cytotoxic (CD4$^-$CD8$^+$) T cells. By using antibodies for pan-myeloid cell marker (CD11b) and neutrophil marker (CADO), the CD5$^-$CD21$^-$ cells were subdivided into three populations, i.e., CD11b$^+$CADO$^+$, CD11b$^+$CADO$^-$, and CD11b$^-$CADO$^-$ (**Fig 1A**). The CD11b$^+$CADO$^+$ population represents neutrophils. The CD11b$^+$CADO$^-$ population but not others included a subpopulation showing high expression of CD14 that represents monocytes (**Fig 1B**). A CD14$^-$ population in the CD11b$^+$CADO$^-$ gate showed higher side scatter (SSC) distribution compared to the CD14$^+$ monocytes and lower forward scatter (FSC) compared to CD11b$^+$CADO$^+$ neutrophils (**S1 Fig**), which represents characteristics of eosinophils [16]. Our data also demonstrated that the B cell population expressed higher level of MHC class II molecule (MHC-II) compared to neutrophils (**Fig 1B**). Interestingly, canine T cells and a subset of monocytes [17] also expressed

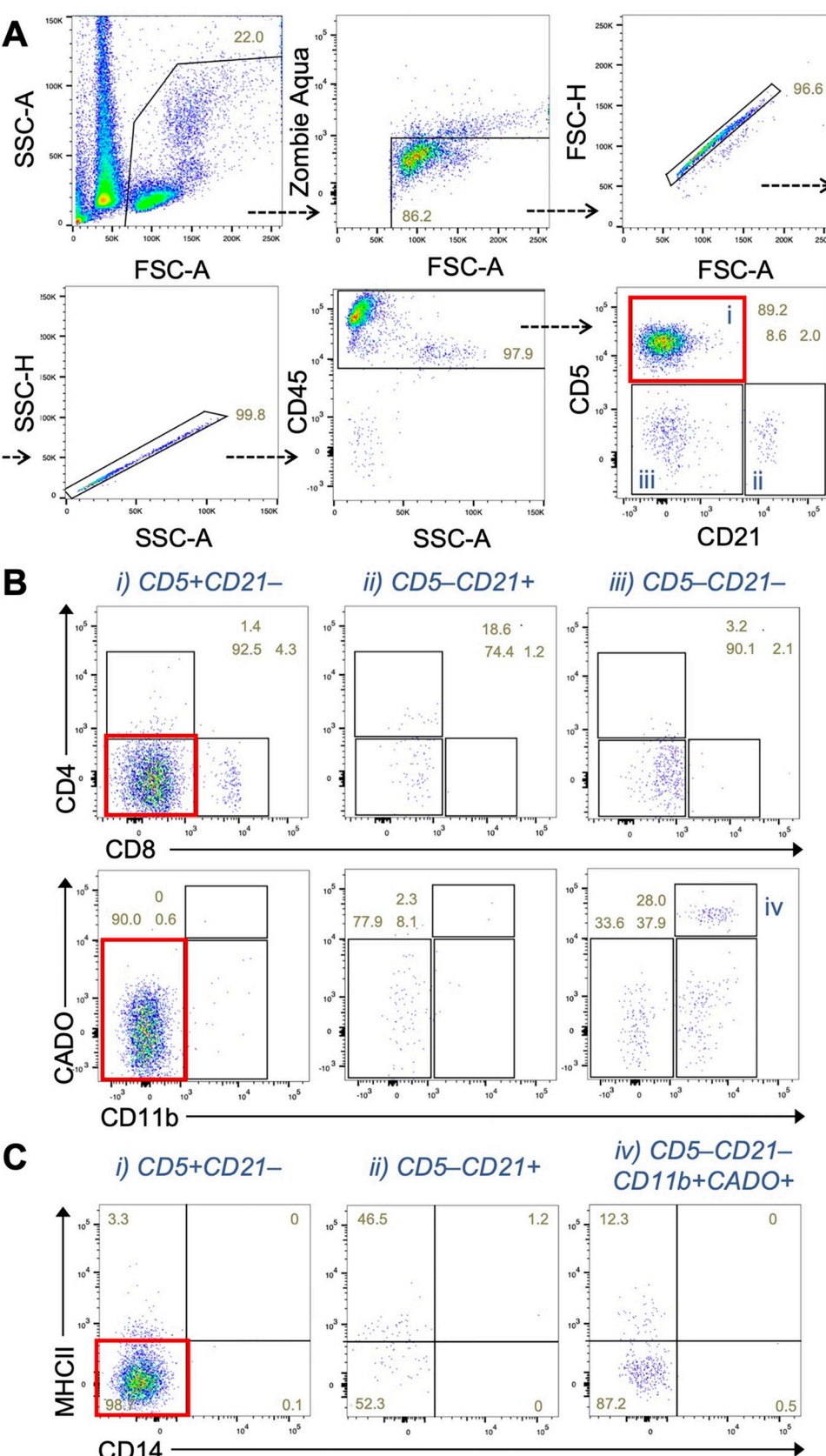

**Fig 2. Sample from a patient suffering from T-cell leukemia shows a uniform expansion of a CD5$^+$CD21$^-$ cell population that does not express CD4 or CD8 in peripheral blood.** (A) Dot plots showing CD5 and CD21 expression by CD45$^+$ cells in peripheral blood of a patient dog. A red rectangle shows the expansion of T cell (CD5$^+$CD21$^-$) population. (B) Expression of CD4 and CD8, or CD11b and neutrophil antigen (CADO) in the indicated immune cell population gated in A. Red rectangles show the absence of CD4, CD8, CD11b and neutrophil antigen expression in the CD5$^+$CD21$^-$ T cell population. (C) Expression of CD14 and MHC class II in the indicated immune cell population gated in A and B.

MHC-II. The subpopulation of CD14$^+$ monocytes defined by low expression of MHC-II might include monocytic myeloid derived suppressor cells [18]. Consistent with previous reports [19], most of neutrophils and a subset of monocytes expressed CD4 at comparable level with T cells (Fig 1C). We confirmed that most of CD5$^+$CD21$^-$ T cells are negative for CD11b and CADO and that majority of CD5$^-$CD21$^+$ B cells are characterized as CD4$^-$CD8$^-$ (S2A Fig), suggesting the appropriateness of threshold used for the CD11b/CADO and CD4/CD8 gating. However, these plots identified a CD11b$^+$CADO$^+$ population within CD5$^-$CD21$^+$ gate, which also express CD4 but not MHC-II (S2B Fig). Since the CD5$^-$CD21$^+$CD11b$^+$CADO$^+$ cells showed higher SSC value compared to CD5$^-$CD21$^+$CD11b$^+$CADO$^+$ cells (S2B Fig), these results suggest the contamination of a neutrophil subset within the CD5$^-$CD21$^+$ gate. The data also indicates that the 9-color panel enables to identify specific immune cell types within a mixed population. Although such contamination within CD5$^-$CD21$^+$ gate does not occur frequently, it is important to examine the expression of markers in each population when analyzing tumor samples (as represented below).

We found essentially the same results in blood samples from dogs presenting with diseases other than cancer (such as chronic enteropathy or endocrine disorders, n = 4). In these samples, the mean percentage in CD45$^+$ leukocytes was 3.3±1.1% in CD4$^+$ T cell, 3.7±1.0% in CD8$^+$ T cell, 2.8±0.6% in B cell, 58.2±8.2% in neutrophil, and 21.4±6.0% in monocyte (10.9±5.9% in CD14$^+$ monocyte). CD4$^+$T/CD8$^+$T ratio and CD8$^+$ T/neutrophil were 0.9±0.1 and 0.08±0.03, respectively. The data indicates that our 9-color flow cytometry panel is sufficient to characterize ranges for frequencies of major immune cell types across different patient cohorts with data acquired from a single panel (in one tube). This is not achievable by commonly used single- or triple-color flow cytometry.

## Application to suspected leukemia samples

To investigate if the 9-color flow cytometry panel can detect neoplastic leukocytes, a peripheral blood sample were collected from a dog with suspected blood cancer based on high white blood cell counts on hematology analysis and analyzed by the panel.

In this sample, we found a uniform expansion (89% in CD45$^+$ cells) of CD5$^+$CD21$^-$ cells (Fig 2A), which represents a T cell leukemia. Interestingly, our panel identified that majority of the CD5$^+$CD21$^-$ cells did not express CD4 or CD8 (Fig 2B). The CD5$^+$CD21$^-$ cells were negative for myeloid cell markers including CADO, CD14 or MHC-II (Fig 2B and 2C). Thus the 9-color flow cytometry panel is applicable for cell populations with neoplastic phenotype in the blood. Although diagnosis of blood cancers is easily done also using 2 to 4 color panels, using 9-color panel is advantageous in detecting less represented subsets in a mixed population such as monocytes.

## Application to lymph node fine-needle biopsy samples

We also examined the ability of the panel to detect neoplastic cells in the lymph node (n = 20). To this end, we analyzed a fine-needle aspirate collected from a dog with suspected lymphoma,

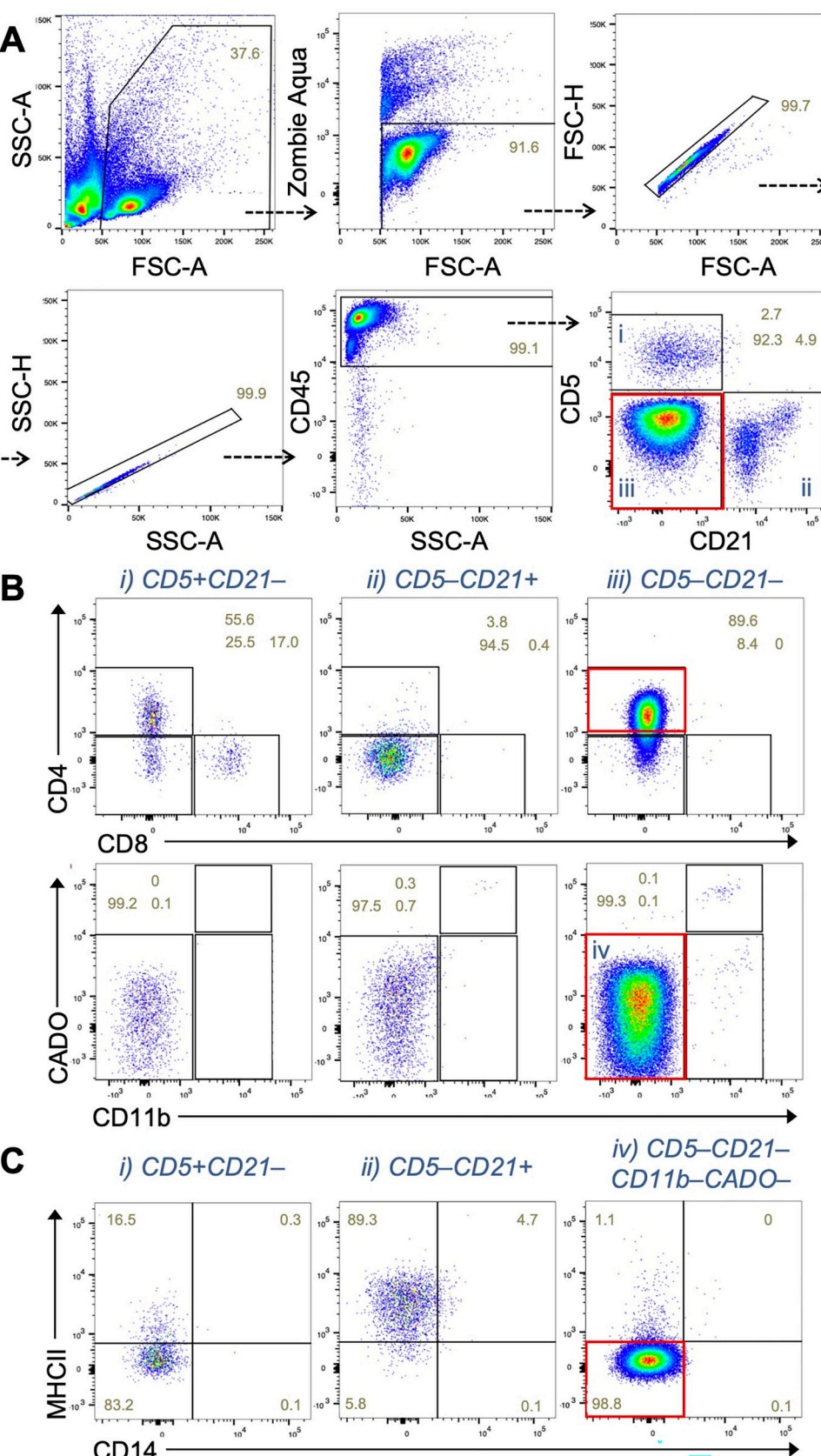

**Fig 3. Predominant expansion of a CD5⁻CD21⁻ cell population that express CD4 but not myeloid cell markers in the lymph node. (A)** Dot plots showing CD5 and CD21 expression by CD45⁺ cells in lymph node samples from a patient dog. A red rectangle shows the expansion of CD5⁻CD21⁻ population. **(B)** Expression of CD4 and CD8, or CD11b and neutrophil antigen (CADO) in the indicated immune cell population gated in A. Red rectangles show expression of CD4 and the lack of myeloid cell markers (CD11b and CADO) in the CD5⁻CD21⁻ population. **(C)** Expression of CD14 and MHC class II in the indicated immune cell population gated in A and B. A red rectangle shows the lack of MHC-II and CD14 expression.

based on clinical presentation of peripheral lymphadenopathy. In this suspected cancer case, we found a predominant population in a CD5⁻CD21⁻ gate (**Fig 3A**). Majority of the CD5⁻CD21⁻ cells expressed CD4 but not myeloid cell markers (i.e., CD11b, CADO, and CD14) or MHC-II (**Fig 3B and 3C**), suggesting that these cells are of T-cell origin.

We also analyzed another sample from an enlarged lymph node in a dog whose blood was examined in Fig 1. In this case, we found the expansion (86% in CD45⁺ cells) of CD5⁻CD21⁺ cells (**Fig 4A**) compared to a lymph node from a dog presenting with non-neoplastic diseases where B cells (CD5⁻CD21⁺) represent 25% of CD45⁺ cells (**S3 Fig**). The CD5⁻CD21⁺ cells did not express CD11b, neutrophil marker CADO, or CD4 while these cells expressed MHC-II (**Fig 4B and 4C**).

These results indicate that the 9-colour flow cytometry panel can distinguish neoplastic cells of B- and T-cell origin in the canine lymph node. Although we selected CD45 positive leukocytes in these examples, CD45 negative neoplastic blood cells such as CD45⁻ T-zone lymphoma can also be characterized by the same panel (**S4 Fig**).

## Application to solid tumor dissociation samples

We further investigated the applicability of the 9-color flow cytometry panel to detect immune cell cells in solid tumor samples (n = 5). Single cell suspensions prepared from a canine melanoma were clearly separated into CD45⁺ (leukocytes) and CD45⁻ (presumably cancer cells) populations. The CD45⁺ cells were further subdivided into CD5⁺CD21⁻ and CD5⁻CD21⁻ populations whereas CD5⁻CD21⁺ B cell population was barely found in this sample (**Fig 5A**). The CD5⁺CD21⁻ population included helper (CD4⁺CD8⁻) and cytotoxic (CD4⁻CD8⁺) T cells and were negative for CD11b or neutrophil antigen (**Fig 5B**). In contrast, CD5⁻CD21⁻ population expressed CD11b and/or neutrophil antigen and was thereby subdivided into CD11b⁺CADO⁺ neutrophils and CD11b⁺CADO⁻ monocytes/ macrophages (**Fig 5B**). Additionally, a subset of cells expressed CD8, but was negative for CD5 or CD21. Given a recent report [20], these cells may represent a subset of NK cells, which further expands the capability of the assay described here. The panel could detect subsets of monocytes expressing CD14 and MHC-II or CD4 and demonstrated that neutrophils in the tumor were characterized as CD14⁻MHCII⁻CD4⁺ (**Fig 5C**). We confirmed that the CD45⁻ population (non-hematopoietic cells) did not express these immune cell markers (**S5 Fig**). These results demonstrate that the 9-color flow cytometry panel is useful to distinct major immune cell types in the tumor (i.e., CD4⁺ T, CD8⁺ T, B, neutrophils, and subsets of monocytes/macrophages).

## Discussion

In veterinary research field, flow cytometry employing a dual or triple staining panel has been used to diagnose canine lymphomas [11, 21–23]. Although this approach has an advantage in accessibility due to the limited requirement of configuration of flow cytometer, its main disadvantage is inability to study more detailed features of gated cells or to distinguish subtypes of immune cells. Although a general diagnostic panel uses about 11 antibodies, fluorochrome conjugated with the antibodies are overlapping and thus 3–4 color staining combinations are

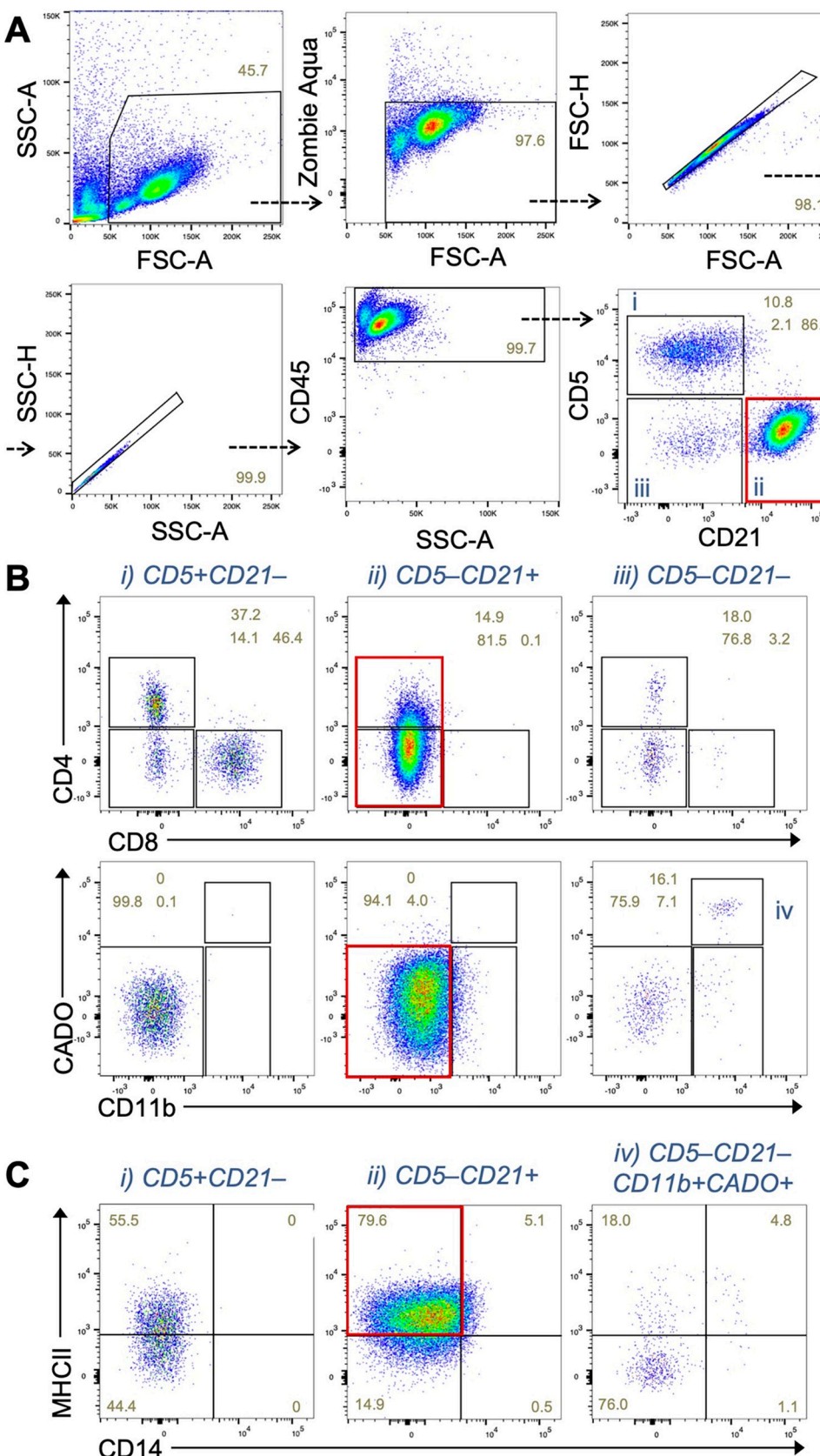

**Fig 4. Predominant expansion of a CD5−CD21+ cell population that express MHC-II in the lymph node. (A)** Dot plots showing CD5 and CD21 expression by CD45+ cells in lymph node samples from a patient dog shown in Fig 1. A red rectangle shows the expansion of B cell (CD5−CD21+) population. **(B)** Expression of CD4 and CD8, or CD11b and neutrophil antigen (CADO) in the indicated immune cell population gated in A. Red rectangles show no expression of CD4 or myeloid cell markers (CD11b and CADO) in the CD5−CD21+ population. **(C)** Expression of CD14 and MHC class II in the indicated immune cell population gated in A and B. A red rectangle shows expression of MHC-II.

required. Therefore, several studies have employed multi-color flow cytometry panels, which enables simultaneous detection of different immune cells in one tube. However, these studies concentrate mostly on the status of T cell populations in healthy dogs [11, 21–23]. Pantelyushin and colleagues have recently described a more extensive flow cytometry panel including markers for lymphocytes (CD3, CD5, CD4, CD8, CD22, CD25, FoxP3) and myeloid cells (CD14 and MHC-II). To our knowledge, this panel utilizes the highest number of markers by combining extracellular and intracellular staining. However, this panel mainly focuses on blood samples collected from healthy dogs and was not validated in tissue samples or neoplastic samples [24]. This is particularly important in translational studies focusing on immunotherapy for blood-derived cancers such as lymphoma and leukemia, where aberrant expression of markers is frequently seen in neoplastic cells. Thorough identification of myeloid cell populations as well as cytotoxic T cells in tumor samples is also important to investigate immunotherapy using canine cancer models since distinct populations of myeloid cells can be associated with impaired recruitment and cytotoxicity of T cells [25].

Herein we described minimum number of markers to distinguish T-cells, B-cells, neutrophils and monocytes/macrophages in canine cancer samples. From a comparative perspective, all the antibodies described for this single panel along with their targets have a direct counterpart in human and mice, allowing for use in translational studies. For the ease of use and reduction of non-specific staining, our panel does not include intracellular marker proteins that require fixation and permeabilization for detection. Our panel includes CD11b and CADO as myeloid cell markers in addition to CD14 and MHC-II, which enables more precise detection of neutrophils and monocyte/macrophage subsets in canine tumor microenvironment. It also allows to determine a ratio of certain immune cell populations that is reported to predict a better prognosis, e.g., high CD8+/CD4+ T cell ratio, high neutrophil/T cell ratio.

We also demonstrated that our panel allows identification of uniformly expanding (i.e., neoplastic) cells in blood and lymph node biopsy samples as well as their origin, and aberrant marker expression in specific cell types if any. For diagnosis of lymphomas, however, cytologic evaluation (e.g., size and morphology of aberrantly expanded cells) is important even if a 9-color panel is utilized. As it has been reported [26], cell size measured via FSC properties by flow cytometry is useful to distinguish B-cell lymphoma cells that are larger than B cells in normal/reactive lymph node (**S6 Fig**). In case of diagnostic fine-needle aspirates, the advantage of using a 9-colour panel, compared to a standard used 2 to 4 color panel, is that only a single tube is sufficient for diagnosis, which is particularly important if low numbers of cells are aspirated. The disadvantage is the required availability of a more advanced, multi-laser, multi-filter flow cytometry system.

The panel currently contains 9 different colors, allowing for further addition of targets of interest in addition to the minimum antibody combination required to detect major tumor infiltrating immune cells. For example, addition of antibodies for immune checkpoint receptors/ligands (e.g., PD-1/PD-L1) and/or T cell exhaustion markers (e.g., TIM3) would be possible to determine T cell status. Given results from human samples [27–29], addition of antibodies for CD11c and CD123 (markers for conventional and plasmacytoid dendritic cells) and/or Siglec-8 (a marker for eosinophil) would enable more detailed discrimination of tumor

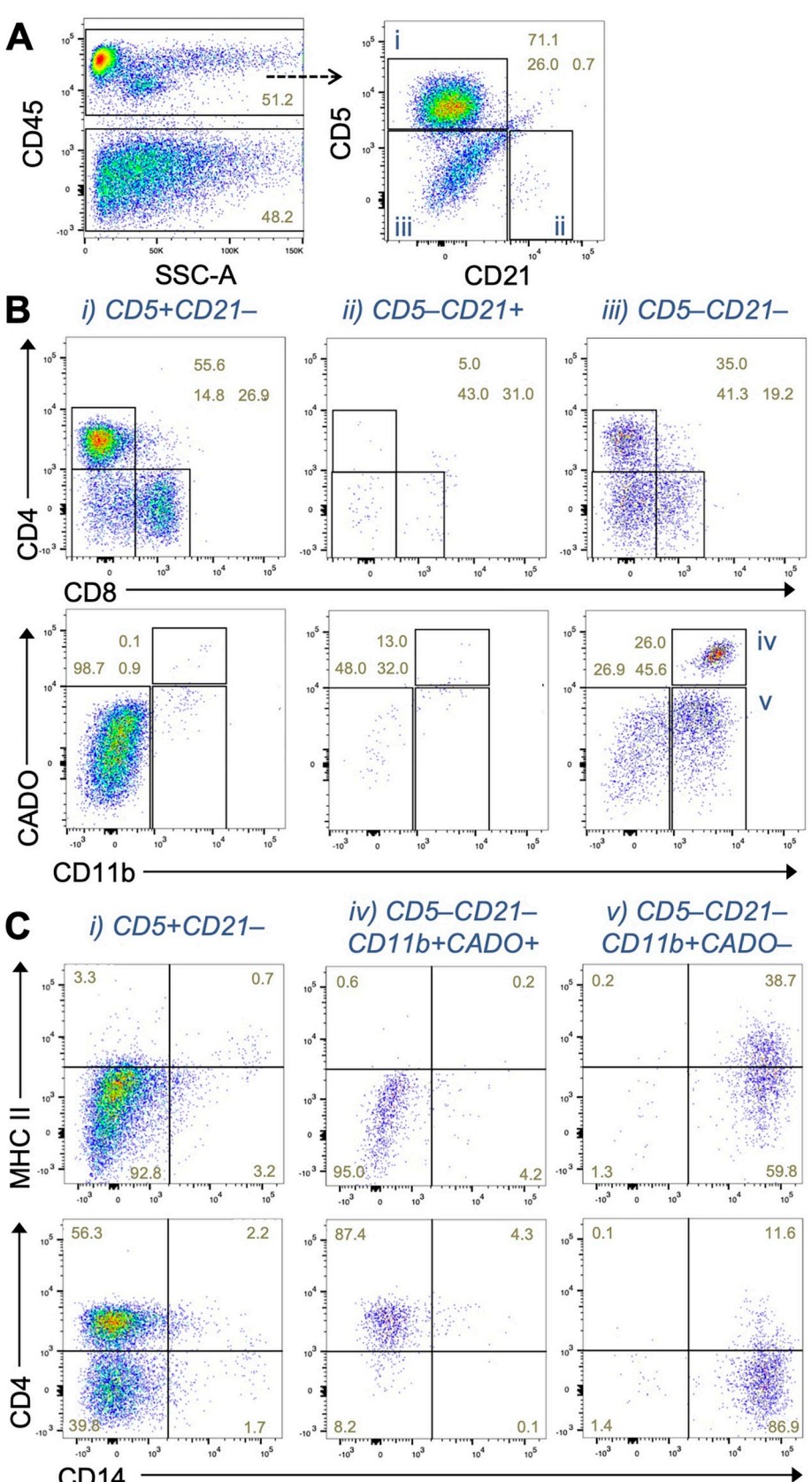

**Fig 5. Immune cell profiling of a melanoma sample. (A)** Dot plots showing CD5 and CD21 expression by CD45[+] cells in a single cell suspension from a canine melanoma. FSC/SSC plots are shown in S5 Fig. **(B)** Expression of CD4 and CD8, or CD11b and neutrophil antigen (CADO) in the indicated immune cell population gated in A. **(C)** Expression of CD14 and MHC class II in the indicated immune cell population gated in A and B.

infiltrating myeloid cells. Although anti-dog CD11c antibody is available for dogs, other antibodies need to be developed or evaluated to check for potential cross reactivity with canine samples. The flexibility of the panel allows it also to be used for leukemia phenotyping through addition of anti-canine CD34 antibody that are labeled by researchers using commercially available antibody conjugation kits with fluorochromes not used in the panel such as PE/Cy5, PE-Texas Red.

In summary, we have described a validated method for multi-color phenotyping of various immune cells in canines. The method provides great advantages for fast and robust characterization of specific cell subpopulations and for the detection of minor/aberrant subsets within a mixed population, which creates novel means for more in-depth studies of canine immune responses.

## Supporting information

**S1 File. Step-by-step protocol, also available on protocols.io.**
(PDF)

**S1 Fig. Forward and side scatter of CD1113[+] myeloid populations. (A)** Expression of CD11b/CADO and CD14/MHCII in non-T/B populations in peripheral blood from dogs shown in Fig 1A. **(B)** Forward (FSC-A) and side (SSC-A) scatter distribution of myeloid cell populations gated in A. Red rectangles suggest that this panel can distinguish cells with eosinophilic properties (CD11b[+]CADO[-]CD14[-]SSC[high]) from monocytes and neutrophils.
(TIF)

**S2 Fig. Detection of a minor population in B cell gate. (A)** Expression of CD11b/CADO and CD4/CD8 and in T cell, B cell, and non-T/B populations in peripheral blood from dogs shown in Fig 1A. Red rectangles show the existence of a minor population in B cell gate. **(B)** Expression of CD4/CD8 and CD14/MHCII in major (CD11b[-]CADO[-]) and minor (CD11b[+]CADO[+]) populations in the B cell gate (CD5[-]CD21[+]) shown in A. FSC/SSC plots of each population (back gating) are also shown. Data suggest suggests the spill over of a neutrophil subset within the B cell gate.
(TIF)

**S3 Fig. Detection of immune cells in a lymph node from a dog presenting with non-cancer diseases.** Dot plots show a gating strategy to detect major immune cell types in lymph node. T, T cells (CD5+CD21-); B, B cells (CD5 -CD21+); Neu, neutrophils (CD5-CD21- CD11b +CADO+); Mo, monocytes (CD5-CD21-CD 1 1 b+CADO-).
(TIF)

**S4 Fig. Detection of CD45- neoplastic T cell population.** (A) FSC/SSC plots of a lymph node sample from a patient dog. (B) Dot plots showing CD5 and CD21 expression by CD45[+] and CD45[-] cells in the sample shown in A. (C) Expression of CD4 and CD8, or CD11 b and neutrophil antigen (CADO) in the indicated immune cell population gated in B.
(TIF)

**S5 Fig. Expression of immune cell markers in CD45[-] cells in a melanoma sample.** (A) Dot plots showing CD5 and CD21 expression by CD45[-] cells in a single cell suspension from a

canine melanoma shown in Fig 5. (B) Expression of CD4, CD8, CD11b, neutrophil antigen (CADO), CD14, and MHC-II in the CD5⁻CD21⁻ cell population gated in A.
(TIF)

**S6 Fig. Forward Scatter (FSC) and Side Scatter (SSC) of CD5—CD21+ cells.** Dot plots showing expression of CD5 and CD21 in CD45$^+$ cells (left) and FSC and SSC in CD5⁻CD21$^+$. cells. Samples were collected from enlarged lymph node of a dog (A, also shown in Fig 4) or a dog with non-cancer disease (B, also shown in **S3 Fig**).
(TIF)

## Acknowledgments

We would like to thank the clinicians, residents and nurses of the Hospital for Small Animals for their assistance with sample collection. We also thank Anna Raper for her assistance with flow cytometry.

## Author Contributions

**Conceptualization:** Maciej Parys, Richard J. Mellanby, Takanori Kitamura.

**Data curation:** Maciej Parys, Takanori Kitamura.

**Formal analysis:** Maciej Parys, Takanori Kitamura.

**Funding acquisition:** Maciej Parys, Takanori Kitamura.

**Investigation:** Maciej Parys.

**Methodology:** Maciej Parys, Takanori Kitamura.

**Resources:** Spela Bavcar, Richard J. Mellanby, David Argyle.

**Validation:** Takanori Kitamura.

**Writing – original draft:** Maciej Parys, Spela Bavcar, Richard J. Mellanby, David Argyle, Takanori Kitamura.

**Writing – review & editing:** Takanori Kitamura.

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
