## [Decision Letter · Decision Letter 0]

19 Jul 2022

PONE-D-22-05345

Use of Multi-Color Flow Cytometry for Canine Immune Cell Characterization in Cancer

PLOS ONE

Dear Dr. Kitamura,

Thank you for submitting your manuscript to PLOS ONE. After careful consideration, we feel that it has merit but does not fully meet PLOS ONE’s publication criteria as it currently stands. Therefore, we invite you to submit a revised version of the manuscript that addresses the points raised during the review process.

Your manuscript has been assessed by two expert reviewers, whose comments are appended below. The reviewers have highlighted some areas where additional information or clarification would be beneficial. Please ensure you respond to each point carefully in your response to reviewers document, and modify your manuscript accordingly.

I see that reviewer 2 has requested that you move information on the protocol into the main text. There is no need to do this, as the Lab Protocol format was designed so that the details of the protocol are hosted on protocols.io (with a copy in the supporting information files) as you have done already. 

We look forward to receiving your revised manuscript.

Kind regards,

Joseph Donlan

Editorial Office

PLOS ONE

2. To comply with PLOS ONE submissions requirements, in your Methods section, please provide additional information on the animal research and ensure you have included details on (1) methods of sacrifice, (2) methods of anesthesia and/or analgesia, and (3) efforts to alleviate suffering.

“This work was supported by Canine Welfare Grant from Dogs Trust UK (Project ID 8841383).”

Please state what role the funders took in the study.  If the funders had no role, please state: ""The funders had no role in study design, data collection and analysis, decision to publish, or preparation of the manuscript.

5. Thank you for stating the following in the Funding Section of your manuscript:

“This work was supported by Canine Welfare Grant from Dogs Trust UK (Project ID 8841383).”

“This work was supported by Canine Welfare Grant from Dogs Trust UK (Project ID 8841383).”

6. Thank you for stating the following in your Competing Interests section: 

“Authors have no competing interests to report.”

7. In your Data Availability statement, you have not specified where the minimal data set underlying the results described in your manuscript can be found. PLOS defines a study's minimal data set as the underlying data used to reach the conclusions drawn in the manuscript and any additional data required to replicate the reported study findings in their entirety. All PLOS journals require that the minimal data set be made fully available. For more information about our data policy, please see http://journals.plos.org/plosone/s/data-availability.

Reviewers' comments:

Reviewer's Responses to Questions

**Comments to the Author**

1. Does the manuscript report a protocol which is of utility to the research community and adds value to the published literature?

Reviewer #1: Yes

Reviewer #2: Yes

2. Has the protocol been described in sufficient detail?

Descriptions of methods and reagents contained in the step-by-step protocol should be reported in sufficient detail for another researcher to reproduce all experiments and analyses. The protocol should describe the appropriate controls, sample sizes and replication needed to ensure that the data are robust and reproducible.

Reviewer #1: Yes

Reviewer #2: Partly

3. Does the protocol describe a validated method?

Reviewer #1: Yes

Reviewer #2: Yes

4. If the manuscript contains new data, have the authors made this data fully available?

Reviewer #1: Yes

Reviewer #2: Yes

**5. Is the article presented in an intelligible fashion and written in standard English?**

Reviewer #1: Yes

Reviewer #2: Yes

6. Review Comments to the Author

Reviewer #1: This manuscript is a very straightforward methods paper describing a 9 color/marker flow cytometry panel for characterizing the major canine immune cell subsets including T and B cells, T cell subsets, and myeloid cell subsets. Furthermore, the manuscript demonstrates application of the panel to both blood and tissues, including lymph nodes and tumor lysate. The utility of the panel in canine oncology as well as other disciplines in canine medicine is without question. Methods are well described, and flow cytometry data are well presented. Details of the patient source are sparse for certain figures as mentioned below in Minor comments. Although this manuscript is not the first report describing a multicolor flow cytometry panel with more 2-4 markers for characterizing immune cell subsets in canine blood and tumor tissue in one tube as stated in the abstract, it is the first report of a panel characterizing broad categories of immune cell subsets in one tube (see comments below). Development of additional multiparameter flow cytometry panels for canine immune cells will facilitate canine cancer models that are considered highly relevant for selected human cancers. Therefore, the content of this manuscript is highly relevant to the field of translational animal models for human cancer. One weakness is the absence of data shown for a healthy control using this panel, although data is shown for canine patients with non-cancer disease. However, given the goals of the manuscript as description of new immunodiagnostic tool for canine cancer patients and illustration of the application of the tool to relevant canine disease, this weakness does not preclude the publication of the manuscript. There are minor comments listed below that should be addressed by the authors before this manuscript is accepted for publication.

Minor Comments:

Line 3: "minor" should be modified to a "small but critical"

Lines 34-35; lines 41-45; lines 91-91: The statements "One possible reason could be the lack of established method to isolate and simultaneously detect different immune cell types in neoplastic tissues"; " To our knowledge, this is the first simultaneous immune cell detection panel applicable for solid tumors in dogs ," and "To date majority of studies base their insights into tumor microenvironment on immunohistochemical studies, or small, limited panels targeting single type of cells [12–14]", are somewhat arguable based on a recent publication (E. E. Sparger et al. Frontiers in Veterinary Science, December 2021, Vo. 8, Article 772932). This report describes 7 and 8 color/marker panels (one tube) to characterize T cell subsets in blood compared to T cell subsets analyzed in melanoma cell lysates from the same patients using a 5 color/marker panel for tumor. Data presented revealed differences in blood versus tumor for specific T cell subsets. Although, this recent report focused on T cell subsets (activation, regulatory and functional markers), rather than broad categories of immune cell subsets, the report should at least be cited and addressed in the Introduction. Strengths of the panel described for this manuscript in review, is the characterization of multiple immune cell subsets rather T cells only, in one tube which can be emphasized and contrasted in the Introduction and Discussion.

Line 34: "of established" should be corrected to "of an established."

Line 39: "allows to detect" is better worded as "allows detection of."

Line 57: "barely" is better worded as "rarely."

Line 62: "environment" should be corrected to "environments."

Line 79: "distinct" should be corrected to "distinguish."

Line 87: "method" should be corrected to "methods."

Figure 1: The data shown in Fig. 1 is identical to that shown in the S1 file where the data is stated to represent dogs showing peripheral lymphadenopathy. Accordingly, the Fig. 1 legend should also include this description. Furthermore, this description of the source for this data is too vague but is clarified by the description of the same patient suggesting a B cell lymphoma, in Figure 4 which shows data from a lymph node aspirate. Clarification of the source of data for Fig. 1 should be better described, at least within the Fig. 1 legend.

The MHCII- population within the CD11b+CADO-CD14+ (monocyte) population is interesting and this phenotype is suggestive of monocytic myeloid derived suppressor cells (MDSC) based on this reference (A. E. Mengos et al.; Frontiers in Immunology; May 2019; Vol. 10; Article 1147). Mention could be made in the Discussion that this population might include this immunosuppressive subset. Conversely this monocyte phenotype might also contain classical monocytes based on reference #16.

Lines 136-137: The reference (#16) cited in the statement, "Consistent with previous reports [16], most of neutrophils and a subset of monocytes expressed CD4 at comparable level with T cells" is not appropriate as this specific reference does not mention or test CD4 expression in canine monocytes.

Lines 157-164: Description of raw data for immune cell frequencies in 4 dogs presenting with non-cancer-related disease, seems out of place as similar data is not presented for healthy controls or the 12 cancer patients. The goals of this manuscript are to describe a 9 color/marker panel for assessment of immune cell subsets within blood, lymph node, and tumor lysate of canine cancer patients, and to illustrate application of a such a panel to these tissue samples. However, presentation of descriptive statistics for a non-cancer cohort as shown in this paragraph, could be justified as to how the data generated by this panel could be analyzed across different patient cohorts and healthy controls. With this in mind, lines 163-164; "distinguish major immune cell types at once (etc)" could be edited to "characterize ranges for frequencies of major immune cell types across different patient cohorts with data acquired from a single panel (in one tube). This is not achievable by commonly used single- or triple-color flow cytometry."

Lines 167-177; Figure 2: Mention should be made that the data shown in Fig. 2 represents a T cell leukemia in this Results section.

A few additional details are warranted on sampling by lymph node aspiration for assessment by flow cytometry. Details such as cell count yields typically necessary for sufficient analysis or cell acquisition would be very helpful for the readers. Also, were multiple aspirates needed to harvest sufficient cells for staining and acquisition? Were such aspirates attempted on enlarged draining lymph nodes associated with canine patients with oral melanoma? These details were not described in the S1 file.

S3 Figure: The plot for the CD5/CD21 cross-gating of CD45+ cells shows the frequencies of each population but the order of the frequencies as shown in the plot appears to be incorrect. Please clarify.

Figure 5B. No mention is made of the CD8+ population observed within the CD5-CD21- population. Although no definitive identification is possible for this CD8+CD5- population given the markers in the panel, recent reports have shown a subset of canine NK cells to be CD8+ similar to human NK cells. This CD8+CD5- population should at least be mentioned as an observation for the lymph node of this patient and the possible identity of this population as NK cells could be mentioned as well.

S6 Figure legend. In the last line, "(B, also shown in Figure S4)," S4 should be corrected to S3.

Line 282: "Enables to identify" should be edited to "allows identification of."

Line 290: "what" should be corrected to "which" and "number" to "numbers."

Line 303-304: "conjugation kit with fluorochrome" should be corrected to "conjugation kits with fluorochromes."

Reviewer #2: This protocol expands the current capabilities to be of use in the clinic. However the authors have dedicated the major component of the protocol to Supplementary information that would make send to be included in the body of the text. There is no mention of the equipment needed to adapt this protocol in comparison with currently standard equipment in practice. Also from a comparative perspective, it would add value if the authors had provided a comparison of markers for each cell types between human, mouse and dogs.

7. PLOS authors have the option to publish the peer review history of their article (what does this mean?). If published, this will include your full peer review and any attached files.

Reviewer #1: No

Reviewer #2: No

---

## [Author Response · Author response to Decision Letter 0]

26 Sep 2022

Response to Editor: 

1. We have ensured that the formatting requirements are met the journal’s requirements. 

2. The animals used in this study are all client-owned pet animals, and the residual samples collected from them for clinical reasons were used for flow cytometry. Each animal had appropriate analgesia provided, which as selected by clinician in charge and depended on the type of procedure/condition. We have added this information in Materials and Methods. The methods of anaesthesia and analgesia were case dependent and selected at the discretion of the clinician in charge. If needed, we can provide this information. However, adding this case dependent information will make this manuscript significantly longer, without adding much value to the protocol. 

3. Word “signed” was changed to “written” to adhere to the suggested wording

4. We would ask for amending the initial submission to include this funding statement:

“This work was supported by Canine Welfare Grant from Dogs Trust UK (Project ID 8841383) - awarded to MP and TK. The funders had no role in study design, data collection and analysis, decision to publish, or preparation of manuscript.”

5. The funding statement has been removed from the manuscript.

6. The authors have declared that no competing interests exist. 

7. We have presented (and thus shared) dot plots including gates and frequency of each population. No graphs or images are shown. Although sharing these processed data is the standard in the field, we have uploaded raw data (i.e., FCS files) to a public repository site (FLOW Repository): http://flowrepository.org/id/FR-FCM-Z5Q7

Response to Reviewer #1: 

Line 3: "minor" should be modified to a "small but critical"

Requested change has been introduced into the text. 

Lines 34-35; lines 41-45; lines 91-91: The statements "One possible reason could be the lack of established method to isolate and simultaneously detect different immune cell types in neoplastic tissues"; " To our knowledge, this is the first simultaneous immune cell detection panel applicable for solid tumors in dogs ," and "To date majority of studies base their insights into tumor microenvironment on immunohistochemical studies, or small, limited panels targeting single type of cells [12–14]", are somewhat arguable based on a recent publication (E. E. Sparger et al. Frontiers in Veterinary Science, December 2021, Vo. 8, Article 772932). This report describes 7 and 8 color/marker panels (one tube) to characterize T cell subsets in blood compared to T cell subsets analyzed in melanoma cell lysates from the same patients using a 5 color/marker panel for tumor. Data presented revealed differences in blood versus tumor for specific T cell subsets. Although, this recent report focused on T cell subsets (activation, regulatory and functional markers), rather than broad categories of immune cell subsets, the report should at least be cited and addressed in the Introduction. Strengths of the panel described for this manuscript in review, is the characterization of multiple immune cell subsets rather T cells only, in one tube which can be emphasized and contrasted in the Introduction and Discussion.

The sentences in Abstract (lines 34-35 and 41-45 in original manuscript) has been modified to “One possible reason could be that there are hardly any established methods to isolate and simultaneously detect a range of immune cell types in neoplastic tissues. To date only a single manuscript describes characterization of immune cells in canine tumour tissues, concentrating solely on T-cells.”. 

The sentence in Introduction has been modified as following, and the suggested study has been referenced. “A recent study has described a 5-color flow cytometry panel that enables to reveal differences between T cell subsets in the blood and those in melanoma tissues [12]. However, this panel is designed to characterize specific T cell subsets, rather than broad categories of immune cell subsets.”. 

Given the reviewer’s suggestion, we have also modified the last sentnce to “The approach described in this study enables the characterization of multiple immune cell subsets rather T cells only in one tube, which will help to build new insights into tumor microenvironment in translational cross-species studies.”.

Line 34: "of established" should be corrected to "of an established."

Line 39: "allows to detect" is better worded as "allows detection of."

Line 57: "barely" is better worded as "rarely."

Line 62: "environment" should be corrected to "environments."

Line 79: "distinct" should be corrected to "distinguish."

Line 87: "method" should be corrected to "methods."

Thank you for the suggestions. We have made all the requested changes in the text.

Figure 1: The data shown in Fig. 1 is identical to that shown in the S1 file where the data is stated to represent dogs showing peripheral lymphadenopathy. Accordingly, the Fig. 1 legend should also include this description. Furthermore, this description of the source for this data is too vague but is clarified by the description of the same patient suggesting a B cell lymphoma, in Figure 4 which shows data from a lymph node aspirate. Clarification of the source of data for Fig. 1 should be better described, at least within the Fig. 1 legend.

The figure legend was updated to “…Representative dot plots showing gating strategy to detect major immune cell types in peripheral blood of dogs with enlarged lymph node and diagnosed with B-cell lymphoma. …”

The MHCII- population within the CD11b+CADO-CD14+ (monocyte) population is interesting and this phenotype is suggestive of monocytic myeloid derived suppressor cells (MDSC) based on this reference (A. E. Mengos et al.; Frontiers in Immunology; May 2019; Vol. 10; Article 1147). Mention could be made in the Discussion that this population might include this immunosuppressive subset. Conversely this monocyte phenotype might also contain classical monocytes based on reference #16.

We have added the following sentence with the suggested citation. “The subpopulation of CD14+ monocytes defined by low expression of MHC-II might include monocytic myeloid derived suppressor cells [18].” 

Lines 136-137: The reference (#16) cited in the statement, "Consistent with previous reports [16], most of neutrophils and a subset of monocytes expressed CD4 at comparable level with T cells" is not appropriate as this specific reference does not mention or test CD4 expression in canine monocytes.

We apologize for the mistake in citation. We have modified the citation to the appropriate one. 

Lines 157-164: Description of raw data for immune cell frequencies in 4 dogs presenting with non-cancer-related disease, seems out of place as similar data is not presented for healthy controls or the 12 cancer patients. The goals of this manuscript are to describe a 9 color/marker panel for assessment of immune cell subsets within blood, lymph node, and tumor lysate of canine cancer patients, and to illustrate application of a such a panel to these tissue samples. However, presentation of descriptive statistics for a non-cancer cohort as shown in this paragraph, could be justified as to how the data generated by this panel could be analyzed across different patient cohorts and healthy controls. With this in mind, lines 163-164; "distinguish major immune cell types at once (etc)" could be edited to "characterize ranges for frequencies of major immune cell types across different patient cohorts with data acquired from a single panel (in one tube). This is not achievable by commonly used single- or triple-color flow cytometry."

Thank you, we have edited the text as requested. 

Lines 167-177; Figure 2: Mention should be made that the data shown in Fig. 2 represents a T cell leukemia in this Results section.

The change requested have been introduced to the figures’ description. 

A few additional details are warranted on sampling by lymph node aspiration for assessment by flow cytometry. Details such as cell count yields typically necessary for sufficient analysis or cell acquisition would be very helpful for the readers. Also, were multiple aspirates needed to harvest sufficient cells for staining and acquisition? Were such aspirates attempted on enlarged draining lymph nodes associated with canine patients with oral melanoma? These details were not described in the S1 file.

We have introduced changes to the protocol (S1 file) where we described the number of aspirates and how many cells are obtained on majority of aspirates. 

S3 Figure: The plot for the CD5/CD21 cross-gating of CD45+ cells shows the frequencies of each population but the order of the frequencies as shown in the plot appears to be incorrect. Please clarify.

Thank you for pointing this out. We corrected the frequency of cell population in the plot. 

Figure 5B. No mention is made of the CD8+ population observed within the CD5-CD21- population. Although no definitive identification is possible for this CD8+CD5- population given the markers in the panel, recent reports have shown a subset of canine NK cells to be CD8+ similar to human NK cells. This CD8+CD5- population should at least be mentioned as an observation for the lymph node of this patient and the possible identity of this population as NK cells could be mentioned as well.

Thank you for the constructive suggestion. We have added the following text and citation to the text: “Additionally, a subset of cells expressed CD8, but was negative for CD5 or CD21. Given a recent report [20], these cells may represent a subset of NK cells, which further expands the capability of the assay described here.”

S6 Figure legend. In the last line, "(B, also shown in Figure S4)," S4 should be corrected to S3.

Thank you, the figure legend for the figure S6 has been modified. 

Line 282: "Enables to identify" should be edited to "allows identification of."

Line 290: "what" should be corrected to "which" and "number" to "numbers."

Line 303-304: "conjugation kit with fluorochrome" should be corrected to "conjugation kits with fluorochromes."

Thank you. We have edited the pointed words accordingly. 

Response to Reviewer #2: 

This protocol expands the current capabilities to be of use in the clinic. However the authors have dedicated the major component of the protocol to Supplementary information that would make send to be included in the body of the text. 

Thank you for your insights into our manuscript. The protocol is added as supplement as it is a requirement of PLos One protocols. 

There is no mention of the equipment needed to adapt this protocol in comparison with currently standard equipment in practice. 

The protocol requires a 14-colour flow cytometer with four lasers (Violet 405nm, Blue 488nm, Green 561nm and Red 640nm) such as LSR Fortessa II. We have added this information as supplement (S1 file). Thus, this protocol is not applicable for a standard practice-based equipment and should be attempted at specialized laboratories. 

Also from a comparative perspective, it would add value if the authors had provided a comparison of markers for each cell types between human, mouse and dogs.

The markers used in the manuscript are identical between species. We have added the following sentence in the Discussion. ”From comparative perspective, all the antibody combination and its targets have a direct counterpart in human and mice, allowing for use in translational studies.”

---

## [Editor Report · Decision Letter 1]

17 Nov 2022

PONE-D-22-05345R1Use of Multi-Color Flow Cytometry for Canine Immune Cell Characterization in CancerPLOS ONE

Dear Dr. Kitamura,

Thank you for submitting your manuscript to PLOS ONE. After careful consideration, we feel that it has merit but does not fully meet PLOS ONE’s publication criteria as it currently stands. Therefore, we invite you to submit a revised version of the manuscript that addresses the points raised during the editor's review process.  We assess that this revised manuscript has significant merit and has addressed the reviewers' comments for the original submission sufficiently and appropriately. However, some errors in word usage, grammar, and syntax persist in the revised manuscript.  Therefore, the current decision for this resubmission is Minor Revision as required to address the editor's comments provided below.  It is important to note that *I participated as a reviewer for the initial evaluation of this manuscript.*

If applicable, we recommend that you deposit your laboratory protocols in protocols.io to enhance the reproducibility of your results. Protocols.io assigns your protocol its own identifier (DOI) so that it can be cited independently in the future. For instructions see: https://journals.plos.org/plosone/s/submission-guidelines#loc-laboratory-protocols. Additionally, PLOS ONE offers an option for publishing peer-reviewed Lab Protocol articles, which describe protocols hosted on protocols.io. Read more information on sharing protocols at https://plos.org/protocols?utm_medium=editorial-email&utm_source=authorletters&utm_campaign=protocols.  *The editor notes that authors have addressed this issue appropriately in this first resubmission.*

We look forward to receiving your revised manuscript.

Kind regards,

Ellen Elizabeth Sparger, DVM, PhD

Guest Editor

PLOS ONE

Journal Requirements:

Editor Comments to address in revised manuscript:

Line 26 page 2: "Development of novel immunotherapies is currently heavily relying" should be edited to "Current development  novel immunotherapies relies heavily."

Line 33 page 2:  "a" should be inserted before "relatively limited amount."

Lines 39-40 page 2:  "enables to characterize" should be modified to "enables characterization of."

Lines 89-90 page 4: "A recent study has described a 5-color flow cytometry panel that enables to reveal" should be modified to "A recent study has described a 5-color flow cytometry panel that enables identification of."

Line 93 page 5: "In here" should be replaced with "Herein."

Line 96 page 5:  "a" should be inserted before "majority."

Line 110 page 5:  "which as selected by clinician" should be modified to "which was selected by the clinician."

Line 269 page 12:  "a" should be inserted before "general diagnostic."

Lines 285-286 page 13:  "From comparative perspective, all the antibody combination and its targets" should be re-worded to "From a comparative perspective, all the antibodies described for this single panel along with their targets."

Title of the S3 figure should be edited from "in lymph node from dogs" to "in a lymph node from a dog"

S4B figure: the scatter plot showing initial interrogation of CD45- cells with defined populations identified as "iii" and "iv", is presumed to be gated for CD5 and CD21 as shown for the CD45+ cells.  However, the axes/gates are not labelled for CD5 and CD21.  This omission should be addressed
---

## [Author Response · Author response to Decision Letter 1]

18 Nov 2022

Line 26 page 2: "Development of novel immunotherapies is currently heavily relying" should be edited to "Current development novel immunotherapies relies heavily." 

We changed the sentence to "Current development of novel immunotherapies relies heavily." 

Line 33 page 2: "a" should be inserted before "relatively limited amount."

Requested change has been introduced into the text. 

Lines 39-40 page 2: "enables to characterize" should be modified to "enables characterization of." 

Requested change has been introduced into the text. 

Lines 89-90 page 4: "A recent study has described a 5-color flow cytometry panel that enables to reveal" should be modified to "A recent study has described a 5-color flow cytometry panel that enables identification of." 

Requested change has been introduced into the text. 

Line 93 page 5: "In here" should be replaced with "Herein." 

Requested change has been introduced into the text. 

Line 96 page 5: "a" should be inserted before "majority." 

Requested change has been introduced into the text. 

Line 110 page 5: "which as selected by clinician" should be modified to "which was selected by the clinician." 

Requested change has been introduced into the text. 

Line 269 page 12: "a" should be inserted before "general diagnostic." 

Requested change has been introduced into the text. 

Lines 285-286 page 13: "From comparative perspective, all the antibody combination and its targets" should be re- worded to "From a comparative perspective, all the antibodies described for this single panel along with their targets." 

Requested change has been introduced into the text. 

Title of the S3 figure should be edited from "in lymph node from dogs" to "in a lymph node from a dog" 

Requested change has been introduced into the text. 

S4B figure: the scatter plot showing initial interrogation of CD45- cells with defined populations identified as "iii" and "iv", is presumed to be gated for CD5 and CD21 as shown for the CD45+ cells. However, the axes/gates are not labelled for CD5 and CD21. This omission should be addressed 

Thank you for pointing this out. We added the label in the plot.

---

## [Editor Report · Decision Letter 2]

1 Dec 2022

Use of Multi-Color Flow Cytometry for Canine Immune Cell Characterization in Cancer

PONE-D-22-05345R2

Dear Dr. Kitamura,

We’re pleased to inform you that your manuscript has been judged scientifically suitable for publication and will be formally accepted for publication once it meets all outstanding technical requirements.

Kind regards,

Ellen Elizabeth Sparger, DVM, PhD

Guest Editor

PLOS ONE

Additional Editor Comments (optional):

Thank you for your attention to the comments of the previous review of the first revised version of your manuscript " Use of Multi-Color Flow Cytometry for Canine Immune Cell Characterization in Cancer". Suggested edits have been successfully introduced into the current revision (PONE-D-22-05345_R2) and your manuscript is ready and accepted for publication. Congratulations!
---

## [Editor Report · Acceptance letter]

5 Dec 2022

PONE-D-22-05345R2 

Use of Multi-Color Flow Cytometry for Canine Immune Cell Characterization in Cancer 

Dear Dr. Kitamura:

I'm pleased to inform you that your manuscript has been deemed suitable for publication in PLOS ONE. Congratulations! Your manuscript is now with our production department. 

Kind regards, 

on behalf of

Dr. Ellen Elizabeth Sparger 

Guest Editor

PLOS ONE